

# Thermokarst lake change and lake hydrochemistry: A snapshot from the Arctic Coastal Plain of Alaska

Lydia Stolpmann[1,2], Ingmar Nitze[1], Ingeborg Bussmann[3], Benjamin M. Jones[4], Josefine Lenz[1], Hanno Meyer[1], Juliane Wolter[1,5], and Guido Grosse[1,2]

[1]Alfred Wegener Institute Helmholtz Centre for Polar and Marine Research, Permafrost Research Section, Potsdam, Germany
[2]Institute of Geosciences, University of Potsdam, Germany
[3]Alfred Wegener Institute Helmholtz Centre for Polar and Marine Research, Section Shelf Sea System Ecology, Helgoland, Germany
[4]Institute of Northern Engineering, University of Alaska Fairbanks, USA
[5]Institute of Biochemistry and Biology, University of Potsdam, Germany

**Correspondence:** Lydia Stolpmann (lydia.stolpmann@awi.de)

**Abstract.** The rapid climate warming is affecting the Arctic which is rich in aquatic systems. As a result of permafrost thaw, thermokarst lakes and ponds are either shrinking due to lake drainage or expanding due to lake shore erosion. This process in turn mobilizes organic carbon, which is released by permafrost deposits and active layer material that slips into the lake. In this study, we combine hydrochemical measurements and remote sensing data to analyze the influence of lake change processes,

especially lake growth, on lake hydrochemical parameters such as DOC, EC, pH as well as stable oxygen and hydrogen isotopes in the Arctic Coastal Plain. For our entire dataset of 97 water samples from 82 water bodies, we found significantly higher $CH_4$ concentrations in lakes with a floating-ice regime and significantly higher DOC concentrations in lakes with a bedfast-ice regime. We show significantly lower $CH_4$ concentrations in lagoons compared to lakes as a result of an effective $CH_4$ oxidation that increased with a seawater connection. For our detailed lake sampling of two thermokarst lakes, we found a significant

positive correlation for lake shore erosion and DOC for one of the lakes. Our detailed lake sampling approach indicates that the generally shallow thermokarst lakes are overall well mixed and that single hydrochemical samples are representative for the entire lake. Finally, our study confirms that DOC concentrations correlates with lake size, ecoregion type and underlying deposits.

## 1 Introduction

Over the last four decades, the Arctic has been warming up to four times faster compared to the global average (Rantanen et al., 2022). This amplified and rapid warming leads to thawing and degradation of permafrost (Biskaborn et al., 2019; Smith et al., 2022), storing about 30 % of the global soil organic carbon (OC) (Hugelius et al., 2014). The mobilization of previously stored OC is driven by permafrost degradation processes such as active layer thickening or thermokarst. These processes lead to subsequent microbial mineralization of OC into greenhouse gasses (Knoblauch et al., 2018; Heslop et al., 2020) or to the

export as particulate (POC) or dissolved organic carbon (DOC) (Fuchs et al., 2020; Stolpmann et al., 2021, 2022). Thermokarst lakes are generally widespread in Arctic lowlands and are highly dynamic features (Grosse et al., 2013; Jones et al., 2022) as



thermokarst processes may result in expanding or shrinking lake surface areas (Nitze et al., 2018a). Thermokarst lakes are considered an important factor of the carbon balance of permafrost regions through emission of carbon dioxide ($CO_2$) and methane ($CH_4$) and sequestering organic matter in lake sediments (Schneider von Deimling et al., 2015; Walter Anthony et al.,

2018; In 'T Zandt et al., 2020). For some lake types, such as Yedoma lakes, it has been shown that increased shore erosion and lake expansion lead to enhanced production of greenhouse gasses from lake sediments and taliks (Walter Anthony et al., 2016). While these processes enhance the emission of $CO_2$ and $CH_4$ to the atmosphere from Yedoma lakes, it is unclear to what extent this is also the case for the diverse types of non-Yedoma lakes such as in northern Alaska. Recent studies showed that $CH_4$ concentrations in lake waters on the Alaska North Slope are relatively low and dominated by young $CH_4$ that originates mostly

in upper lake sediments (Townsend-Small et al., 2017; Elder et al., 2018). Though such measurements of $CH_4$ concentrations in water samples may miss $CH_4$ ebullition as an additional important source for lake $CH_4$ (Wik et al., 2016; Walter Anthony et al., 2021), it provides a generally important insight into the aquatic bio- and hydrochemistry of permafrost region lakes.

Ice-rich continuous permafrost landscapes in northern Alaska are strongly affected by lake-driven degradation processes, including thermokarst subsidence and lake formation, growth, and drainage as well as the subsequent formation of drained

lake basins (Walvoord and Striegl, 2021; Jones et al., 2022). While the Alaska North Slope belongs to the continuous permafrost zone (Jorgenson et al., 2008) and many lakes are primary thermokarst lakes, other lake types are found such as oxbow lakes or intra-dune lakes that formed primarily due to other processes but nevertheless may experience secondary thermokarst dynamics such as shore erosion and talik formation (Jorgenson and Shur, 2007; Arp and Jones, 2009). The morphometric and hydrological characteristics as well as the change dynamics of lakes there are diverse and broadly linked to surface geological

and permafrost characteristics (Hinkel et al., 2012a, b; Arp et al., 2016; Jones et al., 2017; Nitze et al., 2017; Jones et al., 2020; Lara et al., 2021; Lara and Chipman, 2021; Jones et al., 2023).

The thaw of permafrost in a catchment might have direct biogeochemical consequences on lake systems (Kokelj et al., 2005, 2009; Vonk et al., 2015; Gao et al., 2020; Stolpmann et al., 2022). In Northern Alaska, this includes the alteration of water exchange and runoff dynamics (Arp et al., 2012; Liljedahl et al., 2016) and the mobilization of previously frozen

carbon (Kessler et al., 2012; Vonk et al., 2015). The erosion and transportation of freshly thawed sediment material into streams and lakes generally also leads to transfer of OC, both as POC and DOC (Abbott et al., 2015; Wild et al., 2019). While POC is only partially mineralized over longer time-scales and to a large proportion undergoes re-burial in thermokarst lake sediments (Walter Anthony et al., 2014), DOC is highly labile and easily decomposed and mineralized over very short time scales (Holmes et al., 2008; Vonk et al., 2013; Drake et al., 2015; Mann et al., 2015). Thawing of permafrost often

leads to increasing mobilization of DOC (Drake et al., 2015; Mann et al., 2015; Spencer et al., 2015; Mohammed et al., 2022). Several landscape properties are influencing or regulating the concentration of DOC in surface waters (Stolpmann et al., 2021). In thermokarst-dominated landscapes, the concentration of DOC in surface waters is influenced by the surface area, the hydrological connectivity and the water residence time of a water body (Evans et al., 2017). The process of thermokarst lake formation itself leads to increasing DOC mobilization (Spencer et al., 2015) and is influencing the DOC concentration.

For example, Manasypov et al. (2024) found that DOC concentration decreases from thaw depressions to ponds and lakes and increases again during lake drainage or shrinking both in the discontinuous and continuous permafrost zone of Western



Siberia. In northern Alaska, the influence of lake changes, such as lake shrinking and lake expansion, on lake hydro-chemistry parameters is not yet fully investigated and the direct impact of lake shore erosion on lake hydrochemistry parameters has not been considered yet. In addition to the biogeochemical parameters DOC and $CH_4$, other hydrochemical parameters such
as stable water isotopes, anions and cations, pH, and electrical conductivity may provide useful insights to the interactions between thermokarst lakes and landscapes with their specific permafrost and geological characteristics.

Since 20 % of the northern permafrost region are covered by areas that are rich in lakes and drained lake basins (Olefeldt et al., 2016; Jones et al., 2022), a better understanding of how hydrochemistry parameters of aquatic systems are driven by environmental factors in permafrost landscapes is key to better assess and quantify how ongoing and future permafrost
thaw and lake change might impact lakes and their hydrochemical, biogeochemical, and biological roles in a warming Arctic (Vonk et al., 2015; Tank et al., 2020). Considering the large extent of the permafrost region, a combination of lake water hydrochemistry parameters with remote sensing could be useful to extrapolate field measurements to larger regions. Hence it is important to test if lake characteristics and lake dynamics such as changes in surface area or shore erosion correlate to lake water hydrochemistry such as DOC concentrations.

To investigate how a range of hydrochemical parameters, including lake DOC, might correlate with environmental data derived from remote sensing (lake size, area change, and shore erosion rates) and from ancillary datasets such as surface geology, permafrost, and ecoregion type maps we collected and analyzed hydrochemical data from a large number of thermokarst lakes, ponds, and lagoons on the Alaska North Slope. We aim at improving our understanding of the direct impacts of lake change processes on hydrochemical parameters and aquatic ecosystems in Northern Alaska, as this area is changing rapidly under
current climate warming.

## 2 Study area

Our study area is located in the Arctic Coastal Plain (ACP) in northern Alaska from 70°00' to 70°55' N latitude and 152°32' to 154°27' W longitude. It covers the lowland regions surrounding Teshekpuk Lake and is part of the continuous permafrost zone with a permafrost thickness of up to 410 m (Jorgenson et al., 2008). The surface deposits are characterized by ice-rich
marine and terrestrial sediments of the Late Quaternary (Hinkel et al., 2005; Jorgenson et al., 2008; Kanevskiy et al., 2013) that partially are also rich in soil OC (Fuchs et al., 2019; Bristol et al., 2021). The landscape is dominated by thermokarst lakes, drained lake basins, and lagoons with few remaining upland remnants left (Bergstedt et al., 2021). In the outer ACP north of Teshekpuk Lake, lakes cover 22.5 % and drained lake basins 61.8 % of the area (Jones and Arp, 2015). South of Teshekpuk Lake, in the inner ACP, the surface geology transitions to the Ikpikpuk Sand Sea, an area with extensive arrested
sand dunes, ice-poor permafrost, and sparse tundra vegetation, but also with abundant lakes of non-thermokarst origin. Recent studies showed that some lakes in the Teshekpuk study area experience transitions from a bedfast-ice regime to a floating-ice regime due to long-term trends of lake ice thinning, marking a shift from lakes that still have frozen permafrost underneath to lakes that begin developing taliks (Arp et al., 2016; Engram et al., 2018). The opposite transition from floating-ice to bedfast-ice regime also happened for some lakes due to partial drainage (Jones and Arp, 2015; Engram et al., 2018), highlighting the





overall very dynamic landscape in the study area (Nitze et al., 2017).

**Figure 1.** Overview of the study sites (A) on the Alaskan North Slope (Digital Elevation Model) with sampling locations of sampled lagoons, lakes, polygons, and ponds, and sampling locations at the two focus lakes (B) TLO18_12 and (C) TLO18_13.

Our study area generally is influenced by an arctic climate with a mean annual air temperature of -12 °C, and cool summers and winters with mean temperatures of 12 °C and -32 °C in Prudhoe Bay in July and January, respectively (Shulski and Wendler, 2007). Here, semi-arid conditions are found with a mean annual precipitation of 102 mm. The region is covered by

wetland vegetation, dominated by polygonal tundra with wet graminoids and moss communities in the northern part and wet to moist sedge tussock communities in the southern part (Raynolds et al., 2005).

Apart from broad water sampling of lakes in the study area, we selected two focus lakes for more detailed field sampling around their entire shoreline in summer 2018. Focus lake TLO18_12 is located at 70°34' N and 152°32' W in the northeast of Teshekpuk Lake (Fig. 1B). The lake covers an area of 1,102.5 ha and the surrounding area consists of glacio-marine deposits

with peat and pebbly silt and has a maximum water depth of 2.5 m (Hinkel et al., 2016). Focus lake TLO18_13 (informal



name: Peatball Lake) is located at 70°42' N and 153°55' W in the northwest of Teshekpuk Lake (Fig. 1C), covers an area of 114.6 ha, has a maximum water depth of also 2.5 m, is well-mixed (Lenz et al., 2016), and the wider lake area consists of old marine deposits with peat and pebbly silty sand. TLO18_13 has previously been studied extensively for its change dynamics and depositional characteristics (Arp et al., 2011; Lenz et al., 2016; Fuchs et al., 2019) and its talik geometry (Creighton et al.,
2018; Parsekian et al., 2019; Ohara et al., 2022).

## 3   Methods

### 3.1   Field sampling and sampling analysis

During an Alaska North Slope summer expedition in 2018, we collected 97 water samples from 82 different water bodies. The waters samples were collected in the upper 20 cm of the water body and either from the lake center from the floatplane or from
the lake shore. We classified the water bodies following the definition of lakes and ponds by Muster et al. (2017). According to this definition, ponds have a surface area below 1 ha (Rautio et al., 2011). Accordingly, we collected samples from lakes (n=79), ponds (n=3), polygonal ponds (n=7), and lagoons (n=8). Sampled lagoons were classified after Jenrich et al. (2024) into nearly-closed lagoons (n=4), limited-open lagoons (n=3), and a semi-open lagoon (n=1). For our detailed lake analysis, we collected 7 samples from lake TLO18_12 and 10 samples from lake TLO18_13.

For the hydrochemical analysis, water samples were collected with pre-rinsed 250 ml HDPE bottles and splitted in the field lab. For DOC measurements (n=97 samples), a part of the water sample was filtered with a 0.7 μm pore size syringe glass microfiber filter and filled in 20 ml glass headspace vials. We preserved our DOC samples with 30 % hydrochloric acid. In total, 87 samples for stable isotopes analysis were filled in 30 ml PE narrow neck bottles. EC in microsiemens per centimeter ($\mu S\ cm^{-1}$) and pH value were measured for each sample in the field lab with a WTW MultiLab 540. For anions measurements
of 20 samples, a part of the collected water was filtered with a 0.45 μm pore size syringe through a cellulose acetate membrane filter into 8 ml PE wide mouth bottles. For cations measurements of 19 samples, a part of the collected water was filtered with a 0.45 μm pore size syringe through a cellulose acetate membrane filter into 15 ml centrifuge tubes. Cations samples were preserved with 65 % nitric acid.

For further analysis, our samples were transported to labs of the Alfred Wegener Institute Helmholtz Centre for Polar and
Marine Research in Potsdam and on Helgoland. In Potsdam, we used the Shimadzu TOC-VCPH with the non-purgeable organic carbon (NPOC) method to measure the concentration of DOC in milligrams per liter ($mg\ L^{-1}$) in our water samples using the same approach as in Stolpmann et al. (2021). We used the Dionex DX-320 Ion Chromatography System to measure anions and the Perkin Elmer Optima 8300 DV Spectrometer to measure the cations. In the ISOLAB Facility of AWI Potsdam, we measured stable oxygen and hydrogen isotopes, given as delta values ($\delta^{18}$O, $\delta$D) in per mil difference to Vienna Standard
Mean Ocean Water (‰ vs. V-SMOW) by using the equilibration technique with a Finnigan MAT Delta-S mass spectrometer (Meyer et al., 2000). The external error is better than ±0.1 ‰ for $\delta^{18}$O and ±0.8 ‰ for $\delta$D, respectively.

For a total of 44 lakes and lagoons, water samples were filled bubble free into 120 ml glass bottles, conserved with 0.2 ml of 25 % sulfuric acid and closed with butyl stoppers and aluminum crimps. In the home laboratory, a gas phase of 20 ml nitrogen



was created and the $CH_4$ content determined with a Shimadzu GC 2014 with FID (Bussmann et al., 2021). $CH_4$ concentrations were calculated according to Magen et al. (2014) and given in nanomol per liter (nmol $L^{-1}$).

## 3.2 Lake change

We used the lake change detection dataset by Nitze et al. (2018a, b) based on the methodology of Nitze et al. (2017) to extract lake area in hectares (ha), net lake change (ha), net lake change in percentage (%), lake perimeter in meter (m), eccentricity ratio, lake orientation in degree (°), lake solidity ratio (complexity of shoreline, measured as the ratio of pixels in the lake
to pixels of the convex hull image), and elevation in meter (m). This dataset covers a 15-year period from 1999-2014 based Landsat remote sensing data. Due to the spatial coverage of Nitze et al. (2018b), this was applied for 70 water bodies of our dataset. The original dataset covers large parts of northern, interior and western Alaska. As the original dataset did not include annual lake shore erosion rates, we calculated this parameter in centimeter per year (cm $yr^{-1}$) for the 70 samples by using net lake change (ha) and lake perimeter (m) with the following equation:

$$\text{erosion rate} = \frac{(\text{net lake change} \times 10,000)/\text{lake perimeter}}{(15 \text{ years} \times 100)} \tag{1}$$

where positive values mean lake expansion and negative values indicate lake shrinking. For both focus lakes TLO18_12 and TLO18_13 with detailed lake water sampling along the lake shore line, we used georeferenced historical aerial imagery at 1 and 2.5 m spatial resolution from 1955 and 2002, respectively, to map their shorelines from both time steps in the ArcGIS 10 [TM] desktop geographic information system. We then calculated the average annual lake shore erosion rates in meters per
year (m $yr^{-1}$) for this 47-year period using the Digital Shoreline Analysis Software (DSAS) version 5 which is available as an add-on tool to ArcGIS (Himmelstoss et al., 2018) following the method for thermokarst lake shore erosion mapping outlined in Jones et al. (2011).

## 3.3 Additional datasets

We used additional datasets to add lake and lake surrounding properties to our analysis about the influence of lake change
processes on lake hydrochemistry.

In particular, we used the Landscape-Level Ecological Mapping of Northern Alaska and Field Site Photography datasets by Jorgenson et al. (2013). Based on this GIS-ready dataset, we extracted for each lake centroid the ecoregion, physiography, lithology, generalized geology, and soil landscape parameters. Mean elevation of all sampled water bodies was extracted from an airborne IfSAR elevation model available for the Western ACP (Jones et al., 2012; Intermap, 2010). For 87 samples in our
dataset, we extracted information of the lake-ice regime in the winter season and classified floating or bedfast lake-ice regime using the dataset by Grunblatt and Atwood (2014). The lake to ocean distance was calculated by measuring the nearest distance of the shoreline to the coastline.





## 3.4 Statistical analysis

For our statistical analysis we used RStudio (version 4.3.2). We used the Kruskal-Wallis-Test to test the presence of significant
differences between groups and the Dunn's Test to determine significant differences between groups. We used the Pearson's
Product Moment to find significant correlations between two parameters.

## 4 Results

### 4.1 Hydrochemistry

#### 4.1.1 Electrical conductivity (EC) and pH

The EC in the sampled water bodies ranges from 6.5 µS cm$^{-1}$ in lakes to 15,680 µS cm$^{-1}$ measured in lagoons. The overall
median is 252.5 µS cm$^{-1}$, while the median for lakes is 256 µS cm$^{-1}$, for ponds 129.1 µS cm$^{-1}$, for polygonal ponds 159.5 µS
cm$^{-1}$, and for lagoons 10,340 µS cm$^{-1}$ (Table 2, Fig. A1-A). The highest EC was found in a classified limited open lagoon.
Our statistical analysis resulted in significant negative correlations between EC and distance to the coast ($p < 0.05$; cor = -0.3)
and significant positive correlations between EC and $\delta^{18}$O ($p < 0.05$; cor = 0.3) and $\delta$D ($p < 0.05$; cor = 0.4) (Table A1).
Furthermore, differences of EC were significant among water body types with significantly higher EC in lagoons compared
to lakes, polygonal ponds, and ponds ($p < 0.05$; Fig. A1-A). In relationship with the surface geology, we found significantly
higher EC in glacio-marine water bodies compared to water bodies located in eolian sand ($p < 0.05$, Fig. A1-B), significantly
higher EC in water bodies located in old marine deposits compared to eolian sand deposits ($p < 0.05$), and significantly higher
EC in coastal waters compared to water bodies in eolian sand deposits ($p < 0.05$). By analyzing EC in different ecoregion
landscapes, we found significantly higher EC in water bodies of the Arctic peaty lowlands and coastal waters compared to the
Arctic sandy lowlands ($p < 0.05$).

The pH values range from 6.3 to 8.3 with a median of 7.9. In total, only 7 samples have slightly acidic pH values below
7 and were mainly measured in samples from polygonal ponds and ponds with documented brown water. Equal values were
found in a bigger lake of 1,100 ha with a peaty shore and detected foam at the shore. Higher and more basic pH values have
185   been measured in samples collected from lakes with clear or greenish water. Our statistical analysis resulted in a significant
negative correlation between pH and DOC concentration ($p < 0.05$; cor = -0.6) and between pH and $\delta^{18}$O and $\delta$D ($p < 0.05$; cor
= -0.2). We compared pH values in different water body types and found significantly higher pH values in lagoons and lakes
compared to polygonal ponds ($p < 0.05$).





**Table 1.** Range and median of hydrochemistry parameters by sampled lagoons, lakes, polygonal ponds, and ponds.

|  |  | **lagoons** | **lakes** | **polygonal ponds** | **ponds** |
|---|---|---|---|---|---|
|  | *n samples/n lakes* | *8/8* | *79/64* | *7/7* | *3/3* |
| DOC [mg L$^{-1}$] | range | 2.8 to 22.3 | 1.9 to 14.9 | 14.9 to 61.8 | 17.9 to 53.2 |
|  | median | 8.2 | 5.2 | 19.5 | 22 |
| EC [μS cm$^{-1}$] | range | 18.7 to 15,680 | 6.5 to 10,380 | 102 to 252 | 115.3 to 182.4 |
|  | median | 10,340 | 256 | 159.5 | 129.1 |
| pH | range | 7.8 to 8.1 | 6.9 to 8.3 | 6.3 to 7.7 | 6.6 to 7.8 |
|  | median | 8 | 7.9 | 7.2 | 7 |
| CH$_4$ [nmol L$^{-1}$] | range | 14 to 146 | 11 to 1,031 | NA | NA |
|  | median | 19 | 217 | NA | NA |
| $\delta^{18}$O [‰ vs. V-SMOW] | range | -11.74 to -5.02 | -14.36 to -8.84 | -14.38 to -9.2 | -12.22 to -11.18 |
|  | median | -10.47 | -11.92 | -11.23 | -11.75 |
| $\delta$D [‰ vs. V-SMOW] | range | -97.3 to -37.7 | -116.4 to -83.52 | -108.53 to -86.05 | -100.31 to -93.4 |
|  | median | -83.6 | -98 | -96.9 | -98.4 |
| d excess | range | -6.2 to 2.4 | -12.83 to 0.6 | -12.43 to 6.5 | -4.4 to 2.54 |
|  | median | -1.82 | -2.2 | -7.1 | -3.84 |

### 4.1.2 DOC

The DOC concentration of water bodies in our dataset has a median of 5.7 mg L$^{-1}$ and ranges from 1.9 to 61.8 mg L$^{-1}$. We found the lowest DOC concentrations in Teshekpuk Lake and lakes south of it. Samples with a DOC concentration lower than 3 mg L$^{-1}$ have been collected from water bodies with very clear water. Samples of highest DOC concentration ranging from 14.9 to 61.8 mg L$^{-1}$ have been collected from polygonal ponds and ponds smaller than 1 ha in surface area nearby our two focus lakes TLO18_12 and TLO18_13. One exception is a large, nearly closed lagoon with a DOC concentration of 22.3 mg L$^{-1}$ and a surface area of 52.2 ha.

Comparing the DOC concentration with other hydrochemical parameters, we found a significant negative correlation between DOC concentration and pH ($p < 0.05$; cor = -0.6). We also found differences according to water body type with significantly higher DOC concentration in polygonal ponds and ponds compared to lakes ($p < 0.05$; Fig. 2A). Furthermore, we detected differences in DOC concentration by geology. In particular, we found significantly higher DOC concentrations in water bodies located in glacio-marine deposits compared to eolian sand deposits ($p < 0.05$), significantly higher water body DOC concentrations in old marine deposits compared to both old fluvial deposits ($p < 0.05$) and eolian sand deposits ($p < 0.05$; Fig. 2B). We also compared the DOC concentrations in water bodies that are characterized either by bedfast or by floating ice coverage in the winter season and found significantly higher concentrations in water bodies with bedfast ice ($p < 0.05$; Fig. 2D).

The sampled lakes of this study are all located in the Arctic tundra. Dividing this area in different ecoregions, such as Arctic peaty lowland and Arctic sandy lowland, Arctic coastal water, and Arctic silty coast, we found significantly higher DOC con-




centrations in water bodies of the Arctic peaty lowlands compared to Arctic sandy lowlands ($p < 0.05$; Fig. C1-A).

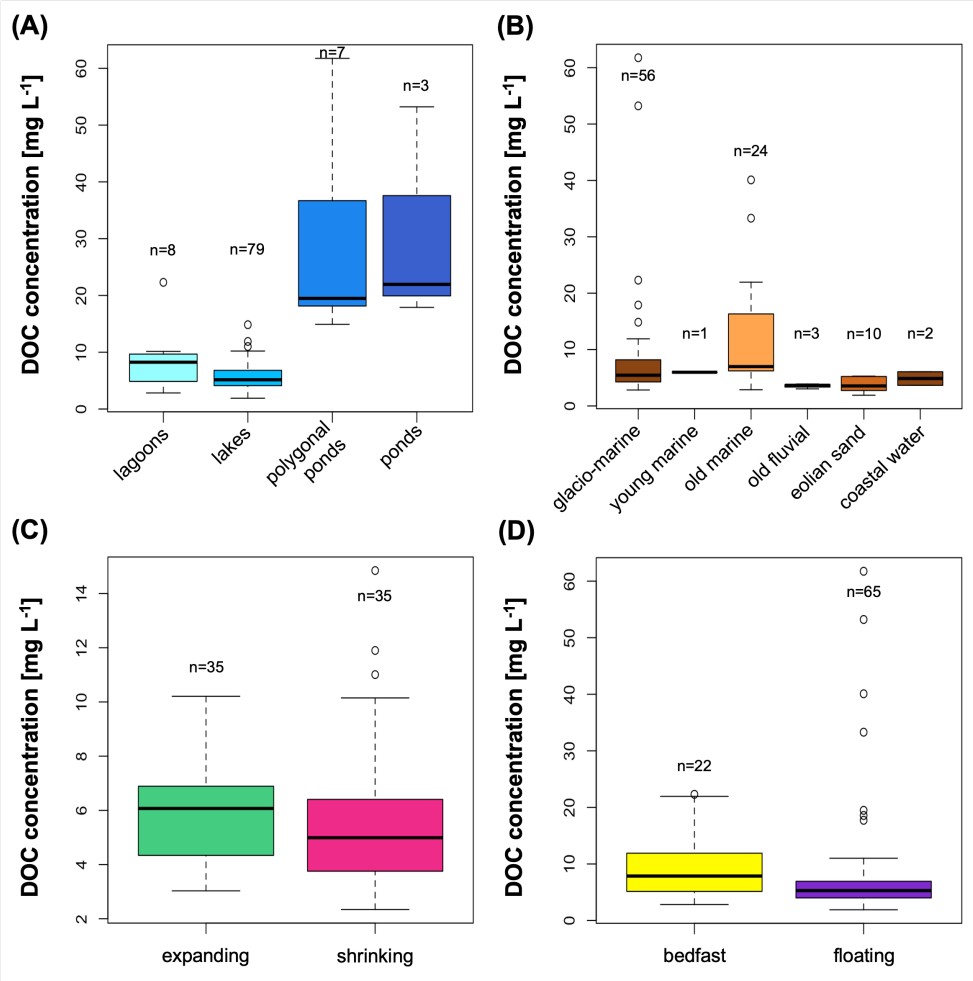

**Figure 2.** Boxplot of measured DOC concentration in collected samples according to (A) classification of surface water type, (B) surface geology after Jorgenson et al. (2013), (C) net lake change after Nitze et al. (2018a, b), and (D) lake-ice regime in the winter season after Grunblatt and Atwood (2014).

### 4.1.3 Methane

The measured water methane concentration ranged from 11 to 1,031 nmol $L^{-1}$, with a median of 130 nmol $L^{-1}$. We found the lowest concentration in samples of Teshekpuk Lake, which is also the largest lake in our dataset. Generally, we found $CH_4$ concentrations below the median of 130 nmol $L^{-1}$ in larger lakes with more than 1,500 ha in surface area. Three lakes with highest $CH_4$ concentrations of more than 900 nmol $L^{-1}$ were observed to have olive brown and partially turbid water with brown and soft mud. Our statistical analyses showed significantly higher $CH_4$ concentrations in lakes compared to lagoons ($p < 0.05$; Fig.





3A). Having a closer look on the $CH_4$ concentrations of sampled lagoons, we found the highest value in a limited open lagoon in a more inland location. We detected no significant difference in $CH_4$ concentration based on the surface geological settings of our study area. However, we found the highest concentrations in water bodies of areas dominated by glacio-marine deposits.

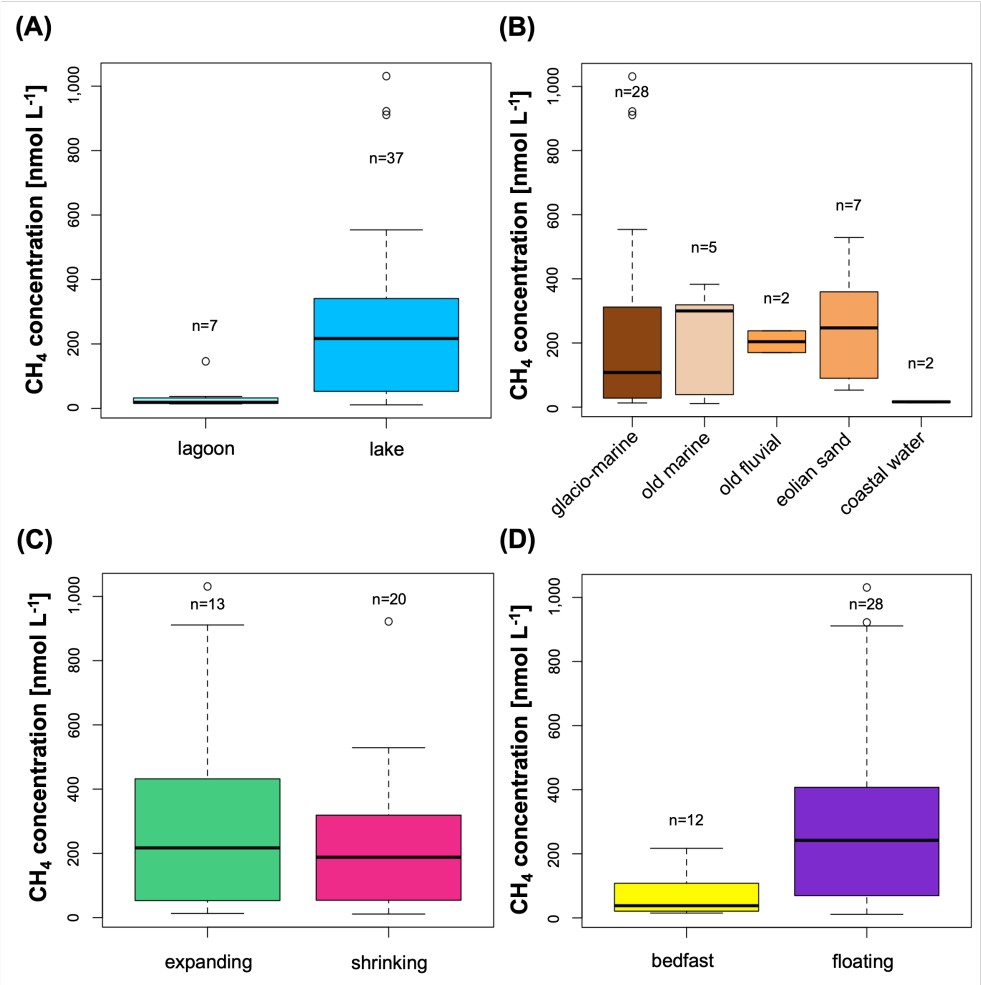

**Figure 3.** Boxplot of measured $CH_4$ concentration in collected samples according to (A) classification of surface water type, (B) surface geology after Jorgenson et al. (2013), (C) net lake change after Nitze et al. (2018a, b), and (D) lake-ice regime in the winter season after Grunblatt and Atwood (2014).

Comparing the $CH_4$ concentrations and the ice coverage in the winter season, we found significantly higher concentrations in water bodies with floating-ice regime ($p < 0.05$; Fig. 3D). While we also found a higher median $CH_4$ concentration of 217 nmol $L^{-1}$ in expanding lakes compared to shrinking lakes with a median of 188 nmol $L^{-1}$ (Fig. 3D), this difference is not statistically significant ($p > 0.05$).



### 4.1.4 Stable oxygen and hydrogen isotopes

The median isotopic composition of all surface water samples is -11.8 ‰ for $\delta^{18}$O and -97.3 ‰ for $\delta$D. The surface water samples are characterized by a median d excess of -2.3 ‰, a slope of 7.3, and an intercept of -10.74 in the $\delta^{18}$O-$\delta$D diagram (Fig. 4). Samples of two rain events during the expedition and one snow sample in the $\delta^{18}$O-$\delta$D diagram (Fig. 4) suggest that the LMWL of Utqiagvik (Barrow) with a slope of 7.5 reflects the present conditions of the study site. Lagoon samples have the heaviest isotopic signature (Fig. 5A) reflecting similar compositions as sea water samples presented in Fig. 4.

For $\delta$D and $\delta^{18}$O we found significant negative correlations with with elevation ($p < 0.05$, cor = -0.4) and distance to the coast ($p < 0.05$, cor = -0.5 and -0.4, respectively) as well as significant positive correlations with EC ($p < 0.05$; cor = 0.4 and 0.3, respectively). More correlations of $\delta^{18}$O and $\delta$D with other parameter can be found in the Appendix Table A1. Comparing water body types, we found a significantly heavier isotopic signature for lagoon samples as for lake samples ($p < 0.05$; Fig. 5A).

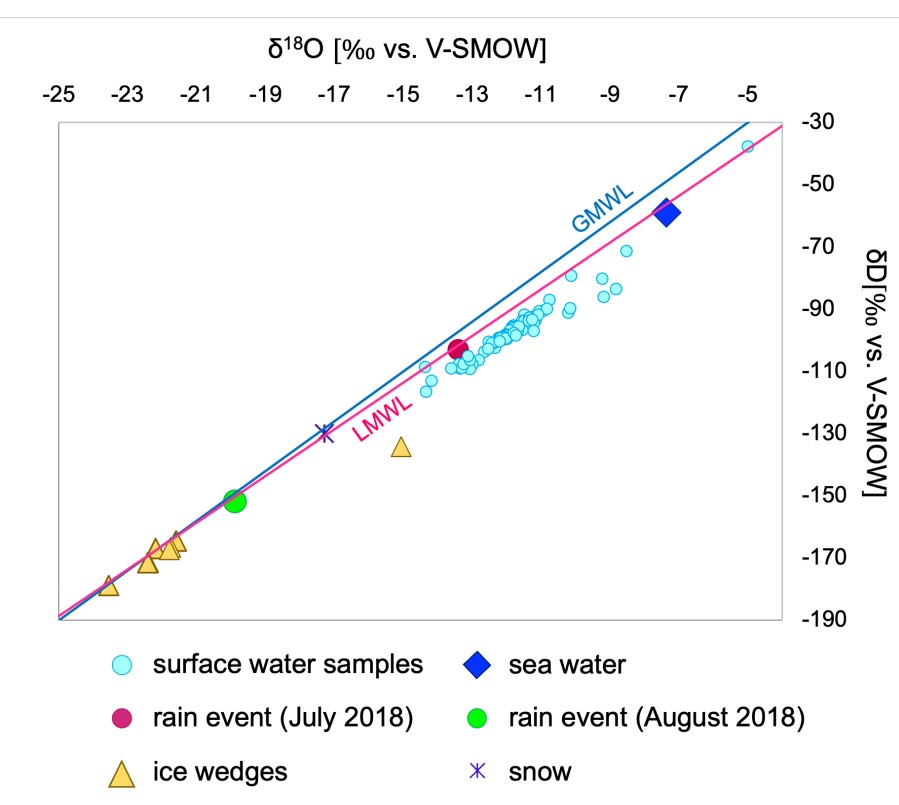

**Figure 4.** $\delta^{18}$O-$\delta$D diagram for all samples of standing water of our dataset. We added ice-wedge samples, sea-water samples, a snow sample and two rain samples collected in summer 2018 on the North Slope, Alaska. GMWL is the Global Meteoric Water Line, whereas LMWL is the Local Meteoric Water Line of Utqiagvik (Barrow), Alaska (Throckmorton et al., 2016).





Lagoon samples have a slope of 8.4 and are close to the GMWL, whereas lake samples have a slope of 6.9, which is close to the LMWL but deviates from the GMWL (Fig. 5A). For the surface geology, we found a significantly heavier isotopic signature for water bodies in glacio-marine deposits compared to old marine, old fluvial, and eolian sand deposits ($p < 0.05$; Fig. 5B). Surface water samples from glacio-marine deposits have a slope of 7.6 that is close to the LMWL, whereas the slope of surface water samples from eolian sand (slope = 6.7), old fluvial (slope = 5.6), and old marine deposits (slope = 5.3),

increasingly deviate from the LMWL (Fig. 5B). We also found a significantly heavier signature in expanding water bodies compared to shrinking water bodies with a slope that is 7.5 and equal to the LMWL for both (Fig. 5C). Furthermore, we found a significantly heavier signature in water bodies with bedfast-ice regime versus floating-ice regime ($p < 0.05$; Fig. 5C). The slope of bedfast-ice regime lakes is 8.3 and close to the GMWL, whereas the slope of floating-ice regime lakes deviates with 6 from the LMWL.

For d excess, we found significant negative correlations with DOC as well as with $\delta$D ($p < 0.05$; cor = -0.3 and -0.4, respectively; Table A1), and a significant positive correlation with pH (p < 0.05; cor = 0.3; Table A1).





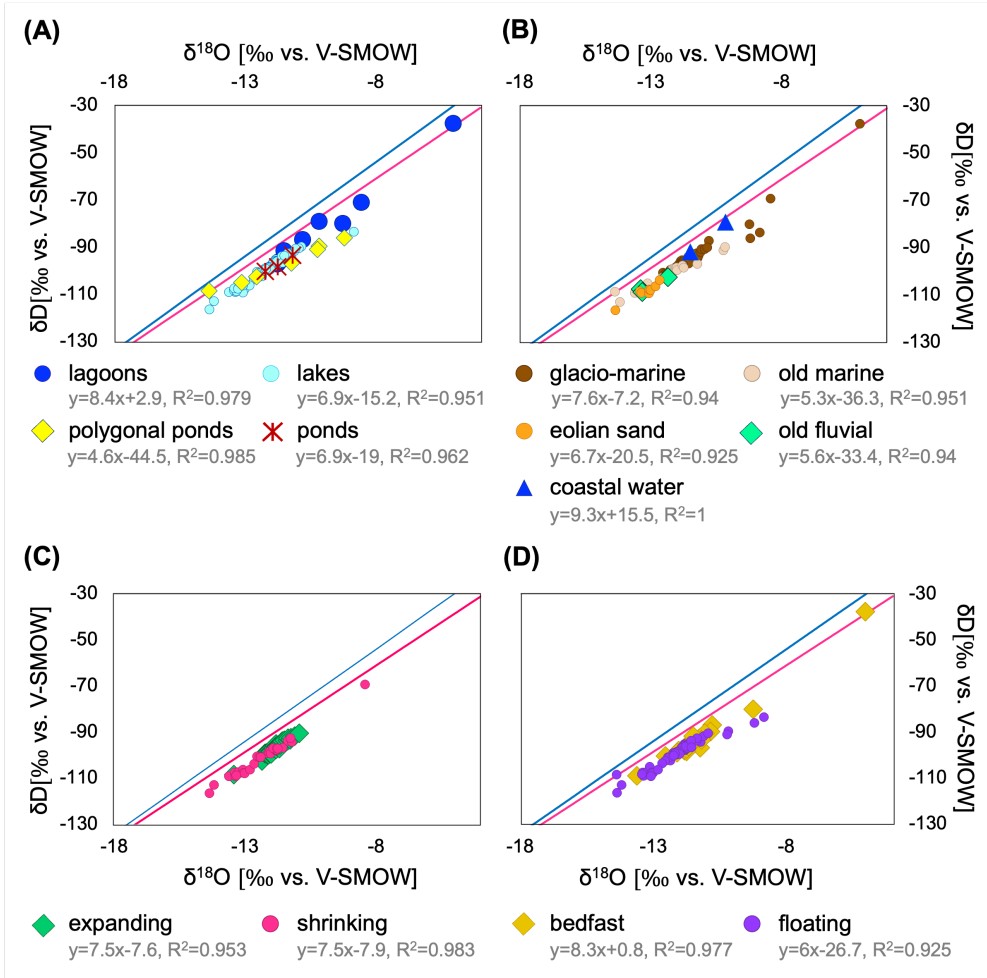

**Figure 5.** $\delta^{18}$O-$\delta$D diagrams of surface water samples of our dataset according to (A) classification of surface water type, (B) surface geology after Jorgenson et al. (2013), (C) net lake change after Nitze et al. (2018a, b), and (D) lake-ice regime in the winter season after Grunblatt and Atwood (2014). GMWL is the Global Meteoric Water Line (blue line), whereas LMWL is the Local Meteoric Water Line (pink line) of Utqiagvik (Barrow), Alaska (Throckmorton et al., 2016).

### 4.1.5 Anions and cations

For 20 lakes of our dataset, we measured aluminum (Al), barium (Ba), calcium (Ca), iron (Fe), potassium (K), magnesium (Mg), manganese (Mn), sodium (Na), phosphorus (P), silicon (Si), and strontium (Sr) for anions. For 19 lakes of our dataset, 250 we measured fluoride, chloride, sulfate, bromide, nitrate, and phosphate for cations. Since values for Al, P, and phosphate were below the detection limits of 100 µg L$^{-1}$, 0.1 mg L$^{-1}$, 0.1 mg L$^{-1}$, respectively, we did not consider these parameters in the following analysis. The same applies for Mn since 18 out of 20 values were also below the detection limit of 20 µg L$^{-1}$. Ranges and medians are displayed in Table 2.





With increasing net lake change (%), we found decreasing concentrations for cations (fluoride, chloride, sulfate, and bromide) with correlation coefficients of -0.8, -0.6, -0.7, and -0.6 ($p < 0.05$; Table A1), respectively. We found increasing concentrations of K, Mg, and Na with increasing gross change rate (cm yr$^{-1}$) with a correlation coefficient of 0.5 for all three anions ($p < 0.05$; Table A1). Furthermore, we found increasing Ba concentrations with increasing distance to coast ($p < 0.05$; cor = 0.7) and decreasing fluoride concentrations with increasing distance to the coast ($p < 0.05$; cor = -0.7).

We present more correlations of anions and cations with other measured hydrochemical parameters in Table A1 and in detail in the Appendix section 'A1 Statistical results for anions and cations'.

**Table 2.** Overall range and median of measured anion and cation concentrations.

|  | range | median |
|---|---|---|
| **Anions** | | |
| Barium [µg L$^{-1}$] | 31-167 | 69.5 |
| Calcium [mg L$^{-1}$] | 10.4-36.1 | 23.4 |
| Iron [µg L$^{-1}$] | 155-324 | 181 |
| Potassium [mg L$^{-1}$] | 0.4-16.5 | 1.5 |
| Magnesium [mg L$^{-1}$] | 1.9-42.9 | 6.5 |
| Sodium [mg L$^{-1}$] | 4-315 | 23.5 |
| Silicon [mg L$^{-1}$] | 0.1-0.2 | 0.2 |
| Strontium [µg L$^{-1}$] | 37.5-270 | 64 |
| **Cations** | | |
| Fluoride [mg L$^{-1}$] | 0.05-0.1 | 0.07 |
| Chloride [mg L$^{-1}$] | 7.8-561 | 44.7 |
| Sulfate [mg L$^{-1}$] | 0.6-64.1 | 1.3 |
| Bromide [mg L$^{-1}$] | 0.05-2.5 | 0.2 |
| Nitrate [mg L$^{-1}$] | 0.27-0.42 | 0.3 |

## 4.2 Lake change

### 4.2.1 Full lake change dataset

In total, 50 lakes of our dataset, including the two focus lakes, were observed in the lake change analysis by Nitze et al. (2018a, b). Of these, we found 19 expanding lakes, including the two focus lakes, and 31 shrinking lakes. We analyzed the relationships between lake change (net change in percentage and hectares, change rate and gross change rate) and different hydrochemical parameters (DOC, CH$_4$, EC, pH, $\delta^{18}$O, $\delta$D, and d excess) and found significant positive correlations between gross change rate and d excess ($p < 0.05$; cor = 0.4; Table A1) as well as the anions K, Mg, and Na ($p < 0.05$; cor = 0.5; Table A1). For the other parameters such as DOC, CH$_4$, EC, or pH no significant correlation was found with the change rates.



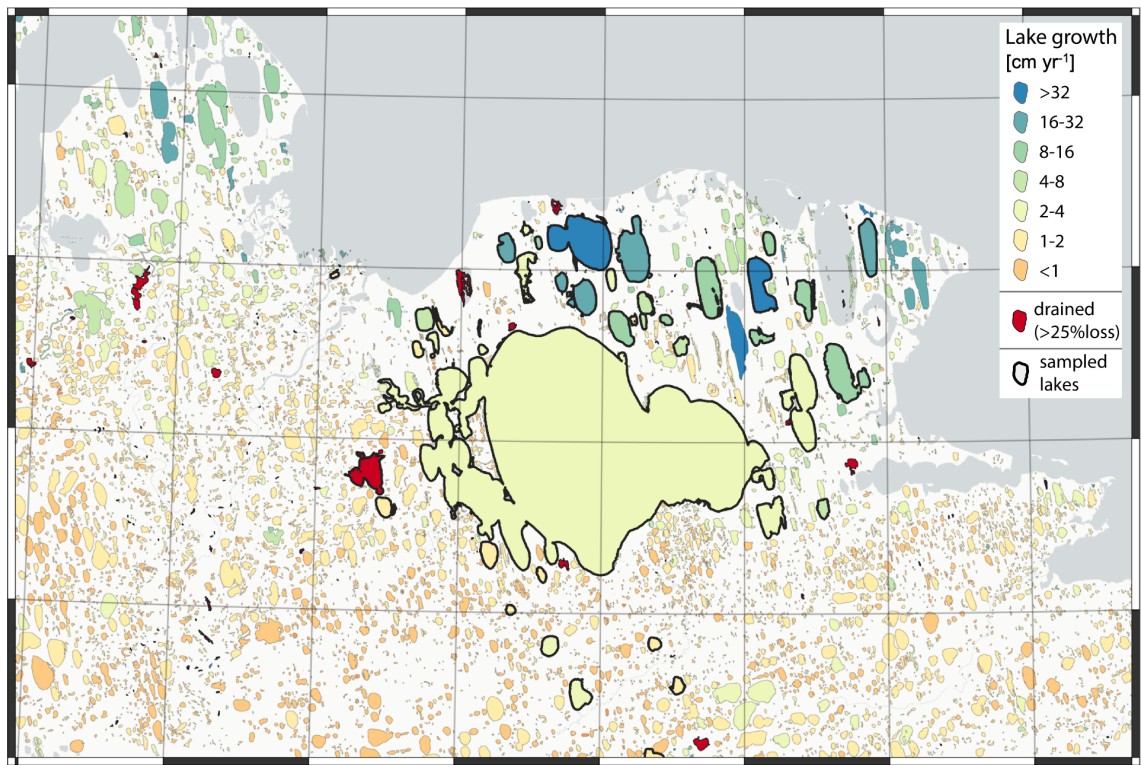

**Figure 6.** Average lake changes in the study area for the period 1999 to 2014 after Nitze et al. (2018a, b). Drained lakes (>25 % area loss) colored in red. Sampled lakes (without lagoons) emphasized by bold outline. Grid size is 90 km latitude and 150 km longitude.

### 4.2.2 Focus lakes

For the two focus lakes, a detailed water sampling along the lake shore was conducted to understand whether hydrochemical parameters differ within these lakes depending on shore characteristics. With a lake size of 1,102.5 ha, lake TLO18_12 is almost ten times bigger than lake TLO18_13 with 114.6 ha and represent rather typical thermokarst lakes in the north of the Teshekpuk Lake region. Both lakes are expanding. In the period from 1999 to 2014, the bigger TLO18_12 had a net lake change of 0.9 % and a calculated mean erosion rate of 38.5 cm yr$^{-1}$, whereas the smaller TLO18_13 had a net lake change of 0.1 % and a calculated erosion rate of 2.5 cm yr$^{-1}$. The surrounding geology of TLO18_12 and TLO18_13 consists of glacio-marine and old marine deposits, respectively. Figure 7 shows detailed annual shore erosion rates from 1955 to 2002 of TLO18_12 and TLO18_13, respectively, and shows that shore erosion occurs in different areas of both lakes. For TLO18_12, annual erosion for the 1955-2002 period is strongest in the southwest corner and on the northern shoreline where maximum annual rates locally exceed 200 cm yr$^{-1}$. For TLO18_13, annual erosion for the 1955-2002 period is strongest in the north and on the eastern shoreline, but maximum annual rates reach 120 cm yr$^{-1}$ locally only.



**Figure 7.** DOC concentration from lake samples from 2018 and annual lake shore erosion rates for focus lakes (A) TLO18_12 and (B) TLO18_13 from 1955 to 2002. Note different map scales for A and B.

We collected 7 water samples from lake TLO18_12 and 10 water samples from lake TLO18_13. We found significantly

285 higher DOC concentrations and EC, and a significantly lighter isotopic composition in TLO18_13 ($p < 0.05$). The DOC concentration in lake TLO18_12 ranges from 4.1 to 5.2 mg L$^{-1}$, and in lake TLO18_13 from 6.1 to 7.5 mg L$^{-1}$. Comparing the DOC concentrations along the shore line of both lakes, we generally found no large variations (Fig. 7; Table B1). The same applies for pH, EC, and the stable hydrogen and oxygen isotopes (Table B1). However, we found a statistically significant but small positive correlation between DOC concentration and annual lake change rate from 1955 to 2002 ($p < 0.05$; cor = 0.7;

290 Table 3B) with higher DOC concentration in areas with a higher change rate for lake TLO18_13 (Fig. 7B). Furthermore, we found a statistically significant but small negative correlation between pH value and annual lake change rate from 1955 to 2002 ($p < 0.05$; cor = -0.7; Table 3B) with lower, more acid pH values in areas with a higher change rate.



**Table 3.** Correlation matrices for samples of the focus lakes (A) TLO18_12 (n=7) and (B) TLO18_13 (n=10) by using the Pearson's Product Moment correlation coefficient between DOC in mg L$^{-1}$, EC in µS cm$^{-1}$, pH, $\delta^{18}$O and $\delta$D in ‰ vs. V-SMOW, d excess, rate for detailed annual lake shore erosion rates in m yr$^{-1}$(from 1955 to 2002), and distance for distance to the coast in km. Significance levels of $p < 0.05$ (in bold) are indicated.

| (A) | | | | | | | | (B) | | | | | | |
|---|---|---|---|---|---|---|---|---|---|---|---|---|---|---|
| | EC | pH | $\delta^{18}$O | $\delta$D | d excess | rate | distance | EC | pH | $\delta^{18}$O | $\delta$D | d excess | rate | distance |
| DOC | -0.3 | **−0.9** | -0.4 | -0.2 | 0.5 | 0.4 | -0.3 | -0.2 | **−0.8** | -0.2 | -0.5 | -0.2 | **0.7** | 0.1 |
| EC | | 0.03 | 0.5 | 0.7 | 0.2 | 0.6 | 0.6 | | -0.1 | -0.2 | 0.02 | 0.2 | -0.06 | 0.6 |
| pH | | | 0.1 | -0.02 | -0.3 | -0.2 | 0.3 | | | 0.3 | 0.5 | 0.06 | **−0.7** | -0.2 |
| $\delta^{18}$O | | | | **0.8** | -0.5 | -0.1 | -0.2 | | | | 0.2 | **−0.7** | 0.05 | -0.4 |
| $\delta$D | | | | | 0.02 | 0.2 | 0.1 | | | | | 0.5 | -0.1 | -0.3 |
| d excess | | | | | | 0.6 | 0.4 | | | | | | -0.1 | 0.1 |
| rate | | | | | | | -0.3 | | | | | | | -0.3 |

## 5   Discussion

### 5.1   Lake DOC correlates with lake size, ecoregion types, and geology

Compared with previous Arctic lake studies, the DOC concentrations of our dataset are slightly higher than in lake samples presented in Stolpmann et al. (2021) but with a median of 5.7 mg L$^{-1}$ generally low. Similar to Shirokova et al. (2013), who investigated DOC in discontinuous permafrost of West Siberia, and the pan-Arctic overview of Stolpmann et al. (2021), our dataset shows increasing DOC concentrations with decreasing lake size. The negative correlation of DOC concentration and lake size is caused by e.g. degradation of vegetation inundated around thermokarst lake margins, low water depth causing lake bottom abrasion by ice coverage in winter, and increased lake sediment respiration of peaty deposits (Shirokova et al., 2013). Analysis of stable hydrogen and oxygen isotopes in our surface water samples show that they are mostly below the LMWL, being influenced by evaporation. The smallest sampled water bodies have the largest deviation below the LMWL, which can be interpreted as a result of higher impacts of evaporation on smaller water bodies. Our results show significantly higher DOC concentrations in lake samples of the Arctic peaty lowlands than in the Arctic sandy lowlands, which likely can be attributed to a generally higher carbon content in the peaty lowland soils (Hugelius et al., 2014). The results of the two investigated focus lakes suggest that single surface water sampling in thermokarst lakes, which are typically shallow (Grosse et al., 2013) and therefore well-mixed, delivers representative values for hydrochemical parameters of the sampled lake. According to our investigations for 2 lakes, we infer that one sample per lake results in representative data for the specific sample day. To reinforce this conclusion, more detailed sampling of lakes in the Arctic is necessary. Seasonal changes however, which we did not investigate in this study, have been shown to be important factors for overall DOC concentrations in thermokarst lakes in particular with the spring freshet and the deepening of the active layer over the course of the summer, both influencing DOC inputs (Manasypov et al., 2015, 2020; Gandois et al., 2021).





## 5.2 Thermokarst lake shore erosion influences lake hydrochemistry

With permafrost degradation, especially abrupt thaw processes such as thermokarst and thermo-erosion, a rapid release of permafrost C is observed (Walter Anthony et al., 2018; Turetsky et al., 2020) and more DOC is being mobilized from a deeper active layer (Wickland et al., 2018). Previous studies suggested that permafrost degradation processes will influence water storage, hydrochemistry, and connectivity of water bodies (Kokelj et al., 2005; Bense et al., 2012). As thermokarst lakes are growing due to erosion of lake shores, surrounding soil and sediment, including from permafrost and the active layer, as well as

organic matter slip into the lake. The transport of organic material into the lake can lead to changes in the lake hydrochemistry, e.g. increasing DOC and particulate OC (POC). Even though, we found only small variations in measured hydrochemical parameters in both focus lakes the results of our detailed analysis of focus lake TLO18_13 confirm a positive correlation with significantly higher DOC concentration directly at the shore in areas of higher detailed annual lake shore erosion rates (Fig. 7; Table 3B; Table B1). In contrast to these results, we found no significant correlation for detailed annual lake shore

erosion rates for focus lake TLO18_12, which is bigger in lake surface area and has generally lower DOC concentrations. Investigations of Textor et al. (2019) in the discontinuous permafrost zone in Central Alaska revealed highly biodegradable DOC in the organic rich active layer. In the continuous permafrost zone, DOC is also highly biodegradable and is especially available for mineralization by microbial activity (Kawahigashi et al., 2004). As DOC is highly biodegradable, this might lead specifically in larger lakes with larger water volumes to the finding that a correlation between DOC concentration with

annual lake change rate cannot be measured anymore. For our entire dataset, we found no big differences between expanding and shrinking lakes in the analyzed hydrochemical parameters. Although we would have expected hydrochemical differences due to the input of permafrost material and meltwater as well as effects of different surface geologies and the role of water evaporation (MacDonald et al., 2021; Wilcox et al., 2023), for example the isotopic composition in our lakes shows very similar values for expanding versus shrinking lakes, except for 3 outliers (Fig. 5C). These outliers are from samples of one lagoon with

typically heavier isotopic composition and two lakes with lighter isotopic composition and have no specific similarities.

## 5.3 DOC and $CH_4$ in bedfast-ice and floating-ice regime lakes

Whereas shallow bedfast-ice regime lakes characteristically freeze to the lake bottom in the winter period, floating-ice regime lakes don't and are characterized by year-round liquid water under the ice (Sellmann et al., 1975). Floating-ice regime lakes are prone to talik development (Ling and Zhang, 2003). These taliks are perennially thawed sediments allowing microbial

activity year-round and are prime environments for $CH_4$ production and release from carbon that was previously frozen in the permafrost (Walter et al., 2007; Heslop et al., 2015). Our results show significantly higher $CH_4$ concentrations in lakes with floating-ice regimes compared to lakes with a bedfast-ice regime (Fig. 3D), which could confirm that taliks underneath floating-ice lakes contribute to higher $CH_4$ concentrations in their water. In northern Alaska on the Arctic coastal plain, Arp et al. (2012) observed a shift from bedfast-ice regime lakes to floating-ice regime lakes since the 1980s. Similar findings were

reported by Surdu et al. (2014) who observed a decreasing number of lakes with a bedfast-ice regime in a subregion of the ACP in Alaska from 1991 to 2011. This lake transformation is critical since it leads to a shift in the water balance and temperature





regime, causing increasing permafrost thaw and degradation beneath lakes (Romanovsky et al., 2010) with floating-ice regime (Arp et al., 2016). The associated talik formation and release of greenhouse gasses is causing a positive feedback (Walter et al., 2006; Shaposhnikova et al., 2023). Considering the rapid climate warming in the Arctic, we can assume an increased release of

greenhouse gasses in these lake-rich areas in the upcoming decades if floating ice-lakes become more abundant. In contrast, we found significantly higher DOC concentrations in lakes with bedfast-ice regime compared to lakes with floating-ice regime , which might be caused by limited water availability for microorganisms in bedfast-ice regime lakes and therefore no or reduced decomposition of DOC occurs (Kurek et al., 2022). Similar results were detected by Bartsch et al. (2017) using a satellite data approach. We found that the stable oxygen and hydrogen isotopes for bedfast-ice and floating-ice regime lakes did not differ

much, except for a few extreme values.

### 5.4   $CH_4$ in lakes and lagoons

We found a high range in $CH_4$ concentration from 11 to 1031 nmol $L^{-1}$ and a median of 158 nmol $L^{-1}$ in our dataset. Our range is similar to findings from Sasaki et al. (2016) who investigated in total 30 lakes along a transect in Alaska, from tundra to mountain and boreal regions. Eleven of their northernmost samples close to Deadhorse ($\geq$ 69° N) ranged from 123 to 1,071

nmol $L^{-1}$. However, their median $CH_4$ concentration of 644 nmol $L^{-1}$ in 2008 and 500 nmol $L^{-1}$ in 2012 is much higher. A study from Samoylov Island in the Lena River Delta (Russia) presented lake $CH_4$ concentrations ranging from 205 to 22,974 nmol $L^{-1}$ and discussed the influence of flooding during the annual spring flood, which transports a large amount of organic material and also carbon, leading to higher $CH_4$ concentrations in their study site (Osudar et al., 2016). A similar effect can be observed in $CH_4$ concentrations from lakes of the Mackenzie River Delta (Canada). Here, Cunada et al. (2018) presented

$CH_4$ concentrations, which are around ten-times higher in samples collected in the end of July and August compared to $CH_4$ concentrations from our dataset. For our study site, the effect of a spring flood does not apply.

The lowest $CH_4$ concentration in our dataset can be assigned to a water sample from Teshekpuk Lake, which is the largest lake in the study area with a surface area of 847.3 km$^2$ (Nitze et al., 2018b). The large size of this thermokarst lake is causing unfavorable conditions for $CH_4$ production. Shirokova et al. (2013) concluded that optimal oxygen supply and missing

phytoplankton debris are causing these conditions in large thermokarst lakes. We also show that production of $CH_4$ might be influenced by erosion processes due to permafrost degradation, resulting in $CH_4$ release and higher $CH_4$ concentrations in expanding lakes (Fig. 3), confirming prior findings of such relationships (Walter et al., 2006; Walter Anthony et al., 2016). Similar to prior findings from the Bykovsky Peninsula in Siberia (Yang et al., 2023), our statistical analysis showed significantly lower $CH_4$ concentrations in thermokarst lagoons (former lakes breached by the sea) compared to thermokarst lakes

that have a median $CH_4$ concentration more than 11-times higher than in lagoons. For lake and lagoon ice core samples from Bykovsky Peninsula, Spangenberg et al. (2021) found similar conditions with a mean $CH_4$ concentration that is 11-times higher in a thermokarst lake than in a neighboring thermokarst lagoon. Lower $CH_4$ concentrations in thermokarst lagoons are a result of lower $CH_4$ production rates and an effective $CH_4$ oxidation, which increases due to the connection with seawater and the introduction of other microbial communities (Yang et al., 2023, 2024). On the other hand, larger lakes would support a greater



loss of $CH_4$ by increased water turbulence and subsequent increased diffusive flux out of the lake, as well as increased $CH_4$
oxidation rates by a better oxygenation of the water (Manasypov et al., 2024).

## 6 Conclusions

Thermokarst lake change caused by permafrost thaw is contributing to the greenhouse effect as part of the global C cycle. In this
study, we investigate hydrochemical parameters – such as DOC and $CH_4$ which are relevant for the emission of the greenhouse
gasses – from a large set of 97 water samples from 82 sampled lagoons, lakes, ponds and polygonal ponds in combination
with remote sensing data for lake change detection, surface geology, permafrost, ecoregion type, and lake ice regimes in the
ACP. For this area around Theshekpuk Lake on the Alaska North Slope, we show that lake shore erosion has an influence on
lake hydrochemical parameters, especially DOC and pH, by releasing organic material from the active layer and permafrost
deposits into the lake. We also show that water bodies that freeze to the bottom in the winter have higher DOC concentrations,
whereas water bodies with a floating-ice regime have significantly higher $CH_4$ concentrations. The latter is especially critical
since previous studies found decreasing lake ice thickness and an increasing abundance in floating ice lakes in many regions
due to rapid winter warming in the Arctic. Comparing lakes and lagoons, we found significantly lower $CH_4$ concentrations in
lagoons. Our detailed sampling of two very different focus thermokarst lakes also suggest that water sampling with a single
sample in thermokarst lakes, which are typically shallow, can deliver spatially representative values for these lakes. We did not
investigate temporal changes in the hydrochemical parameters. Finally, our investigations in the ACP confirm that ecoregion
type as well as geology is influencing the DOC concentration in lake water bodies. Our study improves the understanding of
the direct impacts of lake change processes on hydrochemical parameters and aquatic ecosystems in an area that is changing
rapidly under current climate warming.

*Data availability.* All data from field, remote sensing, and lab measurements from this study can be found in the open-access archive
PANGAEA soon



## Appendix A

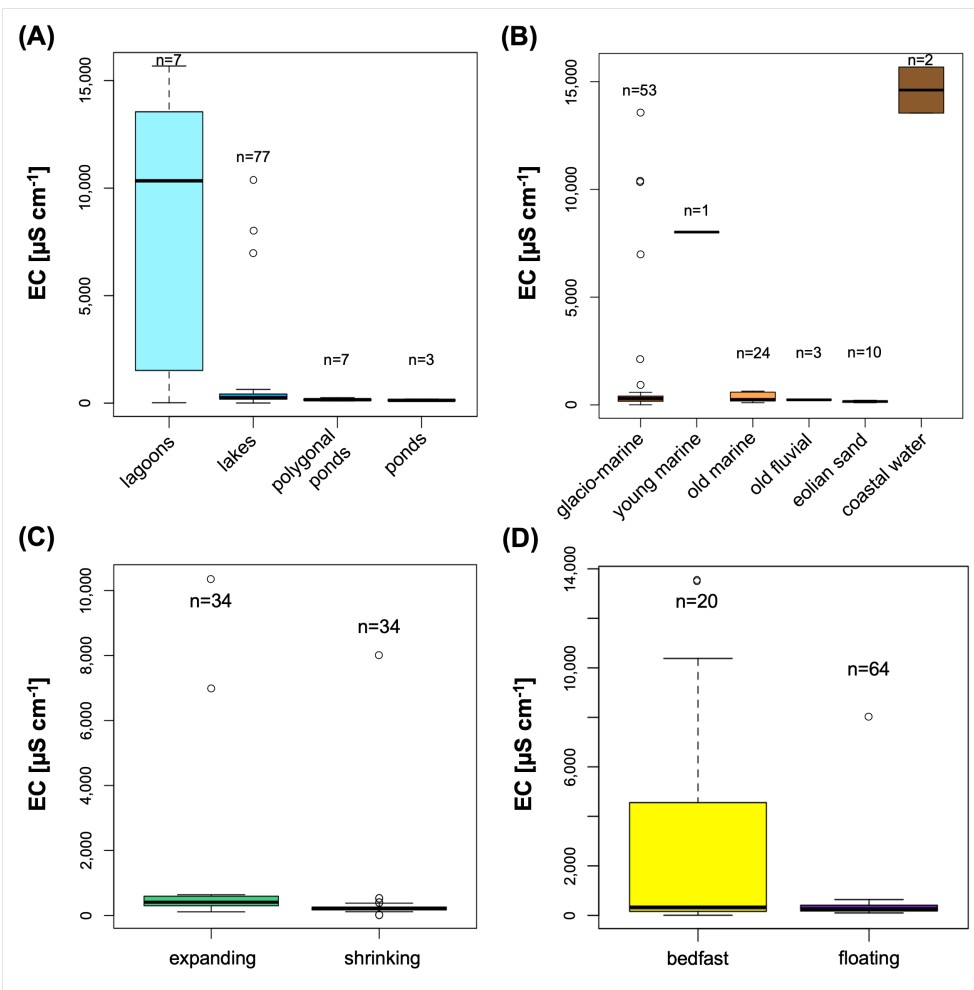

**Figure A1.** Boxplot of measured EC in collected samples according to (A) classification of surface water type, (B) surface geology after Jorgenson et al. (2013), (C) net lake change after Nitze et al. (2018a, b), and (D) lake-ice regime in the winter season after Grunblatt and Atwood (2014).



## A1 Statistical results for anions and cations

For Ba, we found significant positive correlations with pH ($p < 0.05$; cor = 0.7; Table A1) as well as significant negative correlations with DOC ($p < 0.05$; cor = $-0.5$) and CH$_4$ ($p < 0.05$; cor = $-0.6$). For Ca, we found a significant positive correlation with pH ($p < 0.05$; cor = 0.7; Table A1), and significant negative correlations with $\delta^{18}$O ($p < 0.05$; cor = $-0.5$) and $\delta$D ($p < 0.05$; cor = $-0.5$). For K, we found significant positive correlations with DOC ($p < 0.05$; cor = 0.5) and EC ($p < 0.05$; cor = 0.8; Table A1). For Mg, we found significant positive correlations with DOC ($p < 0.05$; cor = 0.6) and EC ($p < 0.05$; cor = 0.9). For Na, we found significant positive correlations with DOC ($p < 0.05$; cor = 0.6) and EC ($p < 0.05$; cor = 0.9). For Si, we found a significant negative correlation with CH$_4$ ($p < 0.05$; cor = $-0.8$). For Sr, we found significant positive correlations with DOC ($p < 0.05$; cor = 0.5) and EC ($p < 0.05$; cor = 0.9). For fluoride, we found significant positive correlations with DOC ($p < 0.05$; cor = 0.6), $\delta^{18}$O ($p < 0.05$; cor = 0.8), and $\delta$D ($p < 0.05$; cor = 0.7). Furthermore, we found significant negative correlations with elevation. For chloride, we found significant positive correlations with DOC ($p < 0.05$; cor = 0.7), EC ($p < 0.05$; cor = 0.5), $\delta^{18}$O ($p < 0.05$; cor = 0.7), and $\delta$D ($p < 0.05$; cor = 0.7). For sulfate, we found significant positive correlations with DOC ($p < 0.05$; cor = 0.6), EC ($p < 0.05$; cor = 0.5), $\delta^{18}$O ($p < 0.05$; cor = 0.7), and $\delta$D ($p < 0.05$; cor = 0.7). For bromide, we found significant positive correlations with DOC ($p < 0.05$; cor = 0.7), EC ($p < 0.05$; cor = 0.5), $\delta^{18}$O ($p < 0.05$; cor = 0.7), and $\delta$D ($p < 0.05$; cor = 0.7). Furthermore, we found significant negative correlations with d excess ($p < 0.05$; cor = $-0.4$). Correlations within anions and cations are shown in Table A1.





**Table A1.** Correlation matrix of entire dataset using the Pearson's Product Moment correlation coefficient between DOC (mg L$^{-1}$), CH$_4$ (nmol L$^{-1}$), EC (µS cm$^{-1}$), pH, $\delta^{18}$O (‰ vs. V-SMOW), $\delta$D (‰ vs. V-SMOW), d excess, barium (µg L$^{-1}$), calcium (mg L$^{-1}$), iron (µg L$^{-1}$), potassium (mg L$^{-1}$), magnesium (mg L$^{-1}$), sodium (mg L$^{-1}$), silicon (mg L$^{-1}$), strontium (µg L$^{-1}$), fluoride (mg L$^{-1}$), chloride (mg L$^{-1}$), sulfate (mg L$^{-1}$), bromide (mg L$^{-1}$), nitrate (mg L$^{-1}$), area (ha) for lake surface area, net lake change (%), net lake change (ha), elevation (m), calculated change rate (cm yr$^{-1}$), and gross change rate (cm yr$^{-1}$) for lakes with measured anion and cation values (n=20). Significance levels of $p < .05$ (in bold) are indicated.

| | CH$_4$ | EC | pH | $\delta^{18}$O | $\delta$D | d excess | Ba | Ca | Fe | K | Mg | Na | Si | Sr | Fluoride | Chloride | Sulfate | Bromide | Nitrate | area | change in % | change in ha | elevation | change rate | change rate gross | distance to coast |
|---|---|---|---|---|---|---|---|---|---|---|---|---|---|---|---|---|---|---|---|---|---|---|---|---|---|---|
| DOC | −.01 | −.002 | **−.6** | .15 | .08 | **−.3** | **−.5** | −.1 | .6 | **.5** | **.6** | **.6** | −.2 | **.5** | **.6** | **.7** | **.6** | **.7** | 0.8 | −.09 | −.2 | .2 | −.2 | .09 | −.1 | −.1 |
| CH$_4$ | | −.2 | −.4 | .08 | .05 | −.3 | **−.6** | −.3 | −.7 | −.1 | −.1 | −.1 | **−.8** | −.2 | −.4 | −.1 | −.2 | −.1 | 0.2 | −.3 | −.02 | .1 | .05 | .05 | −.3 | .06 |
| EC | | | .2 | **.3** | .4 | .1 | .1 | .4 | .2 | **.8** | **.9** | **.9** | .2 | **.9** | .5 | .5 | .5 | .5 | −.5 | −.001 | .08 | .08 | −.2 | .1 | .06 | **−.3** |
| pH | | | | **−.2** | −.2 | **.3** | **.7** | **.7** | .4 | .01 | .04 | .04 | .6 | .2 | .2 | .03 | .01 | .02 | −.9 | .07 | −.03 | −.01 | .2 | −.01 | −.1 | .1 |
| $\delta^{18}$O | | | | | .98 | −.4 | −.2 | −.5 | −.7 | .3 | .4 | .3 | −.1 | .2 | **.8** | **.7** | **.7** | **.7** | .7 | −.1 | −.1 | .2 | −.4 | .2 | .2 | −.4 |
| $\delta$D | | | | | | −.1 | −.2 | −.5 | −.7 | .3 | .4 | .3 | −.1 | .2 | **.7** | **.7** | **.7** | **.7** | .8 | −.1 | −.1 | .2 | −.4 | .2 | .3 | **−.5** |
| d excess | | | | | | | −.1 | .2 | .4 | −.1 | −.1 | −.1 | .5 | −.1 | −.4 | −.5 | −.4 | **−.4** | −.1 | .1 | .03 | −.05 | −.04 | −.03 | .4 | −.05 |
| Ba | | | | | | | | .5 | .6 | −.04 | −.03 | −.02 | .6 | .01 | .01 | −.1 | −.02 | −.01 | −.4 | .03 | −.1 | −.3 | .2 | −.2 | .3 | **.7** |
| Ca | | | | | | | | | .6 | .3 | .3 | .3 | .5 | .5 | **.6** | .3 | .3 | .3 | −.3 | .1 | −.3 | −.3 | .2 | −.3 | .1 | .4 |
| Fe | | | | | | | | | | .4 | .3 | .4 | −.06 | .5 | .5 | .3 | .3 | .3 | NA | −.3 | .5 | **.96** | −.2 | .4 | .1 | −.1 |
| K | | | | | | | | | | | **.98** | **.99** | −.002 | **.96** | .8 | **.99** | **.98** | **.99** | .5 | −.1 | .4 | .3 | −.3 | .4 | **.5** | −.2 |
| Mg | | | | | | | | | | | | **.98** | .2 | **.95** | .9 | **.99** | .9 | **.98** | .7 | −.1 | .3 | .3 | −.4 | .3 | **.5** | −.3 |
| Na | | | | | | | | | | | | | −.1 | **.96** | .8 | **.99** | **.96** | **.99** | .1 | −.1 | .4 | .3 | −.3 | .3 | **.5** | −.2 |
| Si | | | | | | | | | | | | | | .2 | .6 | .04 | .1 | .04 | NA | .3 | .3 | −.4 | .04 | .1 | .6 | .04 |
| Sr | | | | | | | | | | | | | | | .9 | **.96** | .9 | **.96** | .3 | −.1 | .1 | .02 | −.3 | .06 | .5 | −.1 |
| Fluoride | | | | | | | | | | | | | | | | **.8** | **.8** | **.8** | .6 | −.3 | **−.8** | −.02 | **−.6** | −.3 | .2 | **−.7** |
| Chloride | | | | | | | | | | | | | | | | | **.99** | **.99** | .4 | −.1 | **−.6** | .02 | −.2 | −.1 | −.02 | −.3 |
| Sulfate | | | | | | | | | | | | | | | | | | **.99** | .7 | −.1 | **−.7** | .01 | −.2 | −.1 | −.04 | −.3 |
| Bromide | | | | | | | | | | | | | | | | | | | .3 | −.1 | **−.6** | .02 | −.2 | −.1 | −.02 | −.7 |
| Nitrate | | | | | | | | | | | | | | | | | | | | −.5 | .6 | .2 | −.8 | **.9** | **.7** | −.7 |
| area | | | | | | | | | | | | | | | | | | | | | .04 | −.2 | −.04 | −.003 | −.004 | −.02 |
| change in % | | | | | | | | | | | | | | | | | | | | | | **.7** | .06 | **.8** | **.2** | .04 |
| change in ha | | | | | | | | | | | | | | | | | | | | | | | .05 | **.9** | .2 | −.1 |
| elevation | | | | | | | | | | | | | | | | | | | | | | | | .02 | **−.3** | **.7** |
| change rate | | | | | | | | | | | | | | | | | | | | | | | | | .2 | −.1 |
| change rate gross | | | | | | | | | | | | | | | | | | | | | | | | | | −.2 |





**Table B1.** Mean values and standard deviation (SD) for samples collected from the focus lake.

|  | TLO18_12 | | TLO18_13 | |
|  | *n = 7* | | *n = 10* | |
|  | mean | SD | mean | SD |
| --- | --- | --- | --- | --- |
| DOC [mg L$^{-1}$] | 4.6 | 0.5 | 6.6 | 0.5 |
| pH | 7.9 | 0.2 | 8 | 0.1 |
| EC [μS cm$^{-1}$] | 401 | 6.7 | 591.5 | 21.1 |
| $\delta^{18}$O [‰ vs. V-SMOW] | −11.8 | 0.2 | −12.1 | 0.06 |
| $\delta$D [‰ vs. V-SMOW] | −96.8 | 1.3 | −99.2 | 0.4 |
| d excess | −2 | 0.8 | −2.3 | 0.6 |



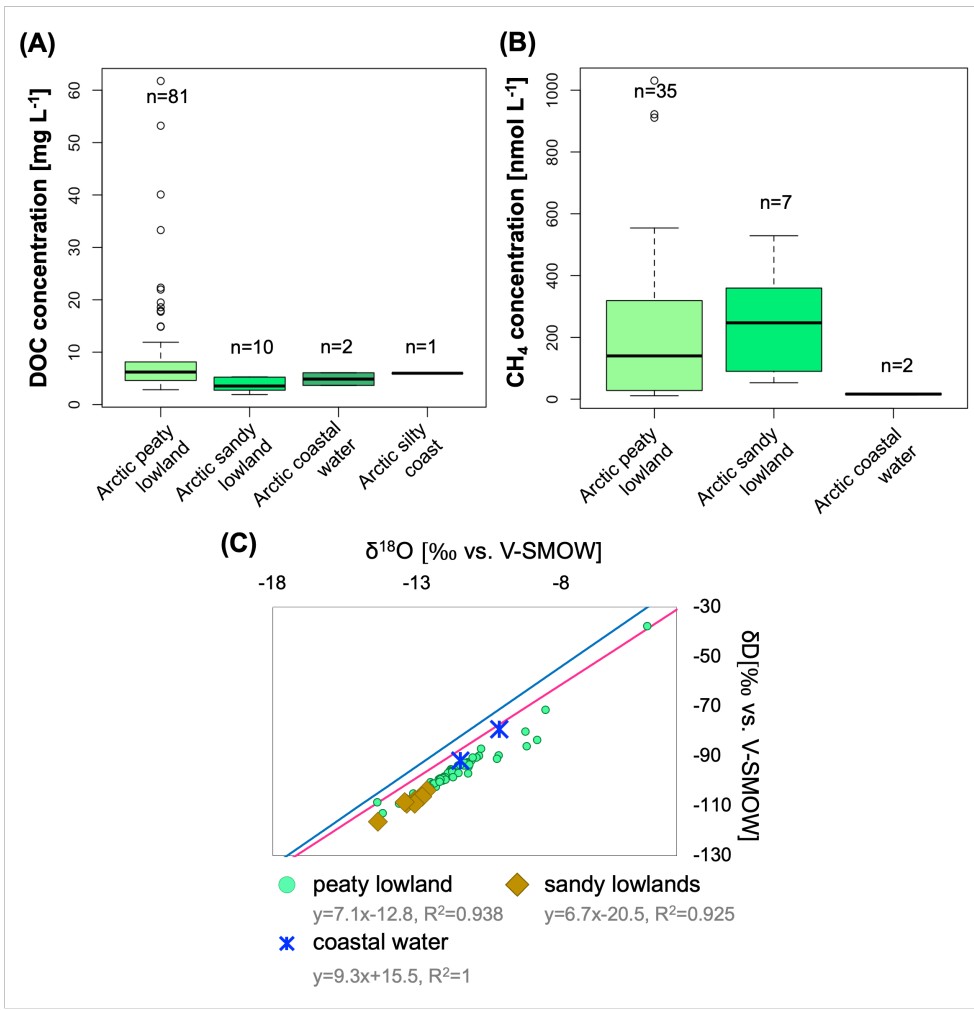

**Figure C1.** Boxplot of (A) measured DOC concentration, (B) $CH_4$ concentrations, and (C) $\delta^{18}O$-$\delta D$ diagram of surface water samples of our dataset according to ecoregion types. GMWL is the Global Meteoric Water Line, whereas LMWL is the Local Meteoric Water Line of Barrow, Alaska (Throckmorton et al., 2016).

*Author contributions.*  LS and GG conceptualized the study. JL, JW, BMJ, and GG conducted the field sampling, field measurements, and field lab sample processing. LS, JL, JW, and IB coordinated and conducted biogeochemical and hydrochemical lab sample processing. HM led the stable isotope sample processing. IN and GG conducted the remote sensing-based lake change analysis. LS led the data analysis and manuscript writing with input from all co-authors.

*Competing interests.*  The authors declare that they have no conflict of interest.

*Acknowledgements.*  We thank the AWI logistics staff for general expedition support and AWI lab staff A. Eulenburg and M. Weiner for help with lab sample processing. We thank S. Schäffler and S. Laboor for their support with GIS and map creation, and M. Angelopoulos for help in the field. We are thankful for the support by the Teshekpuk Lake Observatory which we used as a base camp during the field campaign. We acknowledge the invaluable support for field logistics and sampling for this study by our late floatplane pilot Jim Webster from Webster's Flying Service.

*Financial support.*  LS was funded by a PhD stipend of the Potsdam Graduate School and the Koordinationsbüro für Chancengleichheit of the University of Potsdam, the ERC PETA-CARB project (338335), and the Alfred Wegener Institute Helmholtz Centre for Polar and Marine Research. Field work was funded by AWI base funds and ERC PETA-CARB. JW was funded by the German Research Foundation project QUIC-DRAIN (DFG Research Grant No. WO 2420/2-1). BMJ was supported through U.S. National Science Foundation awards OPP-1806213 and OPP-2336164.

The article processing charges for this open-access publication were covered by the Alfred Wegener Institute, Helmholtz Centre for Polar and Marine Research (AWI).







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
