# Peer review of "Thermokarst lake change and lake hydrochemistry: A snapshot from the Arctic Coastal Plain of Alaska"

_EGUsphere, 2024_

## Author Comment (AC1)

*Point-by-point replies to interactive reviewer comments of*

**Anonymous Referee #1, 12 Dec 2024**

The authors report a novel dataset on hydrochemistry and CH4 regime in lakes from highly remote region, strongly understudied, which has high environmental importance.

This is consistent dataset for large number of variable lakes. It is a pity that CO2 concentrations were not assessed; however, the data are adequately interpreted and the available literature is well captured.

I can recommend moderate revision of this manuscript.

> We are grateful for the positive feedback on our dataset. We agree that direct measurements of $CO_2$ concentrations would have enriched the study. Unfortunately, logistical and analytical constraints during sampling did not allow for any $CO_2$ assessment for our samples. Nevertheless, we aimed to provide robust interpretations using the available data.

**Specific comments**

L171 Reporting median E.C. for this dataset does not make sense – coastal lagoons and thermokarst lakes are incompatible categories

> We thank the reviewer for pointing out the importance of distinguishing between water body types. We agree that combining such contrasting categories can be misleading. In the manuscript, we now changed the sentence to reflect the medians for each water body type (i.e., thermokarst lakes, ponds, and coastal lagoons) as follows:

> *"The median for lakes was 256 µS cm$^{-1}$, for ponds 129.1 µS cm$^{-1}$, for polygonal ponds 159.5 µS cm$^{-1}$, and for lagoons 10,340 µS cm$^{-1}$ (Table 1, Fig. A1-A)."*

L230 edit 'with elevation'

> Thank you very much. We will edit the sentence as follows:

> *"For δD and δ18O we found significant negative correlations with elevation (p < 0.05, cor = -0.4) and distance to the coast (p < 0.05, cor = -0.5 and -0.4, respectively) as well as significant positive correlations with EC (p < 0.05; cor = 0.4 and 0.3, respectively)."*

L249 for cations; L 250 for anions

> Thank you very much. We will change the sentences as follows:

> *"For 20 lakes of our dataset, we measured aluminum (Al), barium (Ba), calcium (Ca), iron (Fe), potassium (K), magnesium (Mg), manganese (Mn), sodium (Na), phosphorus (P), silicon (Si), and strontium (Sr) for cations. For 19 lakes of our dataset, we measured fluoride, chloride, sulfate, bromide, nitrate, and phosphate for anions."*

L326-329 Note also that permafrost thaw an active layer deepening can liberate low molecular weight, potentially highly biodegradable OC from dispersed peat ice (i.e., https://doi.org/10.1016/j.geoderma.2022.116256; DOI: 10.1039/D1EM00547B; https://doi.org/10.1016/j.chemosphere.2020.128953)

We appreciate that the reviewer provided additional literature and notes. We will revise this paragraph as follows:

*"High lability and biodegradability of permafrost and active layer DOC has been demonstrated by many studies such as from the organic rich active layer in the discontinuous permafrost zone of Central Alaska (Textor et al. (2019), permafrost peatlands of western Siberia (Lim et al. 2021; Kuzmina et al., 2023), or the continuous permafrost zone of the Yenisei River region (Kawahigashi et al., 2004). This high DOC biodegradability might lead to a rapid degradation of DOC supplied by shore erosion to larger lakes, and a lack of statistical correlation between DOC concentration with mean annual lake change rates."*

**L341-343 This is important result, that should be stated in the Abstract**

Thank you for highlighting the importance of this result. We accordingly revise the abstract to explicitly include this finding, ensuring it is more visible to readers.

**L379-381 Note that Zabelina et al (2021, doi: 10.1002/lno.11560) also reported a decrease in CH4 concentrations and emissions in large (>100,000 m$^2$) lakes compared to small thaw ponds and lakes (100-10,000 m$^2$).**

Thank you for pointing out the broad study by Zabelina et al. (2021). Their findings generally support our conclusions. We see these results as complementary, highlighting the complexity of GHG dynamics in permafrost regions and the importance of integrating both local process-based and regional assessments. We now added this reference and revised the paragraph as follows:

*"On the other hand, larger lakes would support a greater loss of CH$_4$ by increased water turbulence and subsequent increased diffusive flux out of the lake, as well as increased CH$_4$ oxidation rates by a better oxygenation of the water (Wik et al 2016; Zabelina et al., 2021; Manasypov et al., 2024)."*

**L394-395 This sentence is not necessary for Conclusions**

We agree with the reviewer and will delete it.

---

## Author Comment (AC2)

*Point-by-point replies to interactive reviewer comments of*

**Anonymous Referee #2, 28 Mar 2025**

This manuscript presents hydrochemical data from 82 water bodies in the Arctic coastal plain of Alaska. The authors collected 97 surface water samples and analysed them for concentrations of DOC, CH4, various cations and anions, and water stable isotopes. Two large lakes were studied "in detail", i.e. the authors took seven and ten surface samples, respectively. These hydrochemical data are placed in the context of lake shore erosion rates derived from a previous study by the authors. The topic of the study is indeed highly relevant, as we are witnessing dramatic changes in Arctic landscapes due to rapidly rising temperatures, and we do not know what impact these changes will have on global climate. Furthermore, due to the remoteness of the Arctic, field observations are limited and we urgently need more observations to better understand the current state and improve simulations of future development. The manuscript is well written and the data are clearly presented. On the other hand, much more could have been gained from this study by going beyond standard analyses. A 14C analysis of DOC would have helped to assess whether the DOC originates from surface active layer material or from permafrost deepening, and a d13C analysis of dissolved methane would have helped to assess whether low CH4 concentrations in the lagoons are due to low CH4 production or increased CH4 oxidation, to name only a few examples.

> We appreciate the reviewer's suggestion regarding additional isotopic analyses. We agree that $^{14}$C analysis of DOC and $\delta^{13}$C analysis of methane could provide valuable insights into the sources and age of carbon and would certainly enhance the depth of the study. However, due to the logistical and financial constraints associated with conducting fieldwork in such a remote and challenging location, we were unable to perform these specific analyses during this project and we hope to consider and implement them in future field campaigns.

The authors' aim was 'improving our understanding of the direct impacts of lake change processes on hydrochemical parameters'. However, due to the sampling design, this study does not contribute much to this goal. More than 7-10 surface water samples should have been taken per lake and, more importantly, the samples should have been taken according to the lake shore erosion rates. In particular, at lake TOL18_12, samples were taken in areas with similar shoreline erosion rates, although the rates vary greatly, and data from TLO18_13 indicate a relationship between erosion rate and DOC concentrations.

> We respectfully acknowledge that a more extensive spatial sampling strategy could have provided a more comprehensive picture of the biogeochemical processes. However, due to logistical constraints associated with the remoteness and limited accessibility of the study area, we focused our efforts on capturing a broad regional set of lakes and the most feasible and representative sites for a more detailed sampling based on existing knowledge of the two lakes near Teshekpuk Lake Observatory. We agree that expanding the sampling coverage and also an increase in sampling detail both spatially and temporally would strengthen future studies for this region. While we recognize the spatial limitations, we believe that the findings presented still offer valuable insights and highlight some emerging key patterns that can inform and motivate future, broader-scale investigations.

As it stands, the study mainly confirms what was already known, such as that surface water DOC concentrations in well mixed lakes do not differ very much (which seem obvious, otherwise they would not be well mixed), that DOC concentrations are higher in lakes overlying peaty sediments than overlying sandy sediments, that lakes with year-round unfrozen sediments (taliks) produce more methane than those that freeze to the ground in winter, or that CH$_4$ concentrations in lagoons are lower than in freshwater lakes. Furthermore, some of the data presented are not discussed at

all, such as anion and cation concentrations, or only very briefly, such as water stable isotopes. If these data are not important for the study, I suggest not presenting them.

To improve the manuscript, authors should focus on results that surprised them rather than on results that confirm previous knowledge.

We appreciate the reviewer's perspective and concerns regarding the interpretation and presentation of certain data. For this particular region in the Arctic coastal plain of the Alaska North Slope such data is sparse and has not been widely collected. We therefore strongly believe that this rich hydrochemical exploratory dataset from 82 water bodies warrants analysis and publication. We find it rather encouraging that some of the results align with findings from other regions reported in the existing literature, providing additional context and strengthening our understanding of the processes at play in this specific study area. We included additional data measured, such as anions, cations, and stable isotopes, though not discussed in as much depth as other aspects of the data, to offer a more complete view of the biogeochemical dynamics in the region. This data supports the overarching narrative of the study by providing important chemical signatures that complement measurements we presented. These results were only briefly discussed to indicate their relevance, however, we agree with the reviewer that we should expand a bit on the discussion of these variables in the revised manuscript to better highlight their implications for the overall findings. We respectfully believe that omitting these data would reduce the depth of the study and the ability to make robust comparisons to other published works. To address the reviewer's concern, we will clarify and expand the significance of these variables in the revised manuscript.

Specific comments:

L9: I think taking seven or ten surface water samples from two lakes larger than 100 ha may not be called 'detailed lake sampling'.

We appreciate the reviewer's comment and recognize that the use of the phrase "detailed lake sampling" could be seen as misleading. To clarify, we softened our terminology and now refer to the more intensive sampling conducted at two lakes as follows in the abstract: "We conducted additional subsampling along the shorelines of two thermokarst lakes and found a significant positive correlation for lake shore erosion and DOC for one of these lakes. Our additional lake sub-sampling approach...". We will adjust the wording in other parts of the manuscript accordingly to clarify the additional subsampling at two lakes in contrast to the remaining lakes that were sampled with a single sample per lake for broader regional context.

L78: Refer to Figure 1

Thank you very much. We agree that referring to Figure 1 at this point helps to get a picture of the study area. We will change the sentence as follows:

"Our study area is located in the Arctic Coastal Plain (ACP) in northern Alaska from 70∘00' to 70∘55' N latitude and 152∘32' to 154∘27' W longitude (Fig. 1A)."

L165f: Both tests are used to test for significant differences between groups?

To clarify, we suggest rephrasing the sentence as follows:

"We used the Kruskal-Wallis-Test to detect the presence of significant differences between groups and the post-hoc Dunn's Test to identify which groups differ significantly."

**L172: Table 1**

Thank you very much for bringing this to our attention. We will change the sentence as follows:

"The EC in the sampled water bodies ranges from 6.5 µS cm$^{-1}$ in lakes to 15,680 µS cm$^{-1}$ measured in lagoons. The median for lakes is 256 µS cm$^{-1}$, for ponds 129.1 µS cm$^{-1}$, for polygonal ponds 159.5 µS cm$^{-1}$, and for lagoons 10,340 µS cm$^{-1}$ (Table 1, Fig. A1-A)."

**L248: You are confusing cations with anions (also in Table 2 and the following text). Also, how do you measure phosphorus and phosphate? Phosphorus was probably only measured as phosphate. If elements were below the detection limit, I suggest just mentioning this without writing that they were measured.**

We appreciate the reviewer's attention, and apologize for this obvious oversight. We will ensure that the terminology is consistent and accurate in the revised manuscript.

We thank the reviewer for this comment regarding phosphorus versus phosphate. Indeed, we used ion chromatography (ICS2100 Dionex/ThermoFisherScientific) for the determination of dissolved anions, including Phosphate. We used the inductively coupled plasma optical emission spectrometry (ICP-OES) (Perkin-Elmer Optima 8300DV) for elemental determination of dissolved cations, including Phosphorus.

We thank the reviewer for this comment. Our intention in listing all measured anions and cations — including those below the detection limit — was to maintain transparency and to give a complete account of the analytical scope of the study. We agree that values below the detection limit do not contribute to quantitative interpretation; however, we believe it is still useful for readers to know which elements were analyzed, even if they were not detected in measurable concentrations.

**L324: I do not see that the data from TOL18_12 are in contrast to the data from TOL18_13, since the samples from the shore at TOL18_12 are from sites with similar shore erosion rates, and therefore a dependence between shore erosion rate and DOC concentration cannot be evaluated. Unfortunately, areas with higher shore erosion rates were not sampled at TOL18_12. I would shorten the discussion here, as the limited data and sampling sites give only limited insight into the effect of shore erosion on lake hydrochemistry.**

We agree with the reviewer and removed the wording "In contrast to these results" accordingly. We also added the following statement in the discussion, highlighting the uncertainty based on the still rather low sample number: "Generally, for our detailed sampled lakes the water sample numbers are still low and for a conclusive link between shore erosion rates and DOC more targeted sampling along gradients of different erosion rates and more lakes will be needed."

**L377ff: It should be made clear that these statements about CH4 production and CH4 oxidation refer to the anoxic sediments of the lagoons, not the water columns from which the samples were taken.**

We thank the reviewer for this comment and agree that this should be clarified as follows:

"*Lower CH$_4$ concentrations in thermokarst lagoons are a result of lower CH$_4$ production rates and an effective CH$_4$ oxidation in lagoon sediments, which increases due to the connection with seawater and the introduction of other microbial communities (Yang et al., 2023, 2024; Jenrich et al., 2025).*"

**L388: I have not seen any mention of pH in the discussion.**

We thank the reviewer for pointing this out. It is correct that pH was not specifically discussed in the manuscript, as the values observed were within expected ranges and did not show unusual or site-specific variability that would have warranted further interpretation. However, we agree that for the sake of consistency, parameters mentioned in the conclusion should be addressed in the main text. To address this, we will include a brief note on the pH data in the Discussion, indicating that the pH values were consistent with prior studies and typical ranges for lakes in the region, not exhibiting patterns requiring deeper analysis.